# Microstructure and Interfaces of Ultra-Thin Epitaxial AlN Films Grown by Plasma-Enhanced Atomic Layer Deposition at Relatively Low Temperatures

**Ramasis Goswami [1,*], Syed Qadri [1], Neeraj Nepal [2] and Charles Eddy, Jr. [2]**

[1]  Naval Research Laboratory, Materials Science and Technology Division, 4555 Overlook Ave. SW, Washington, DC 20375, USA; syed.qadri@nrl.navy.mil

[2]  Naval Research Laboratory, Electronics Science & Technology Division, 4555 Overlook Ave. SW, Washington, DC 20375, USA; neeraj.nepal@nrl.navy.mil (N.N.); charles.eddy@nrl.navy.mil (C.E.J.)

*  Correspondence: ramasis.goswami@nrl.navy.mil

**Abstract:** We demonstrate the growth of ultra-thin AlN films on Si (111) and on a GaN/sapphire (0001) substrate using atomic layer epitaxy in the temperature range of 360 to 420 °C. Transmission electron microscopy and X-ray diffraction were used to characterize the interfaces, fine scale microstructure, and the crystalline quality of thin films. Films were deposited epitaxily on Si (111) with a hexagonal structure, while on the GaN/sapphire (0001) substrate, the AlN film is epitaxial and has been deposited in a metastable zinc-blende cubic phase. Transmission electron microscopy reveals that the interface is not sharp, containing an intermixing layer with cubic AlN. We show that the substrate, particularly the strain, plays a major role in dictating the crystal structure of AlN. The strain, estimated in the observed orientation relation, is significantly lower for cubic AlN on hexagonal GaN as compared to the hexagonal AlN on hexagonal GaN. On the Si (111) substrate, on the other hand, the strain in the observed orientation relation is 0.8% for hexagonal AlN, which is substantially lower than the strain estimated for the cubic AlN on Si(111).

**Keywords:** atomic layer deposition; aluminum nitride; gallium nitride; interfaces; transmission electron microscopy

## 1. Introduction

Aluminum nitride is a III-N compound semiconductor with a band gap in the range of 6.01–6.05 eV at room temperature [1]. This semiconducting material is a useful candidate in a number of potential applications in microelectronics because of its relatively high thermal conductivity, high electrical resistivity, low thermal expansion, chemical stability in air up to 1380 °C, and excellent thermal shock resistance. The epitaxial crystalline aluminum nitrides are used to manufacture surface acoustic wave sensors because of the piezoelectric properties of AlN [2,3]. It exhibits both wurtzite and zinc-blende phases as do other III-N compounds. In the present paper, we report on the low-temperature growth of ultra-thin (<20 nm) binary AlN films using atomic layer deposition (ALD) growth processes. The low-temperature growth process allows the elimination of miscibility gaps in ternary III-N semiconductors and reduces extended defect formation due to thermal coefficient of expansion mismatch that affects the quality of the films grown with the conventional higher temperature growth methods. Metal–organic chemical vapor deposition (MOCVD) and molecular beam epitaxy have been used to grow good quality AlN films. However, it involves high temperatures that results in the obstruction of growing strain-free heteroepitaxial film as well as the full stoichiometric range of alloyed III nitrides [2–5]. The miscibility gap in AlN occurs above 500 °C, which can lead to spinodal decomposition during growth at these temperatures. At the present time, the III-nitride films of highest quality are deposited by molecular beam epitaxy and metalorganic chemical vapor depo-

sition methods. It has been reported that temperatures of 900 °C and higher are usually needed to obtain high-quality AlN films by these methods.

The epitaxial growth of AlN has been carried out at high temperatures, usually at temperatures greater than 1300 °C, which enables the bulk diffusion of Al and reduces the density of defects [6,7]. The growth involves different methods at a relatively high temperature, such as epitaxial lateral overgrowth [8–11], pulsed atomic layer epitaxy [12–16], and high-temperature post-growth annealing [17–19]. However, the high-temperature growth has inherent difficulties as it introduces high thermal stresses and buckling due to the thermal expansion mismatch between the substrate and AlN film [20].

To overcome these difficulties, currently, the ALD technique has been largely explored to deposit AlN films at relatively low temperature of 640 °C [21] as the dimensions of the device are reduced and the non-planar complexity is enhanced [22]. However, films grown at reduced temperatures, 1100–1200 °C, contain a high density of point defects [23,24] and threading dislocations [25–28]. These films also exhibit rough surface morphology [26,28]. In addition, the low-temperature ALD growth could yield amorphous film or metastable phases in the film. Recently, ALD growth for AlN has been reported in the temperature range of 150–280 °C using tris (diethylamido) aluminum and hydrazine ($N_2H_4$) or ammonia [29] and in the temperature range of 175–350 °C using trimethyl aluminum (TMA) and hydrazine [30]. The use of highly reactive nitrogen source, such as hydrazine, provides a higher deposition rate per cycle as compared to ammonia. However, the low-temperature deposition of AlN shows a higher level of impurity content, particularly hydrogen [29].

To facilitate epitaxial films at relatively low temperatures, plasma-enhanced ALD (PDALD) processes can be used [30,31]. In this process, films are grown layer-by-layer with adsorption of the group III-containing molecule first, followed independently by adsorption/reaction with the group V atom. This is actually a non-equilibrium growth process that is comprised of two surface-limited half-reactions. Thus, the growth process is a complex one, which uses the metal–organic precursors and reactive radicals from a plasma source. As this process involves surface-mediated chemical reactions, understanding the interface, defects, and chemical composition at the nanoscale and fine-scale microstructure of atomic layer epitaxy grown film would be crucial to control the properties of these films. Not much work has been done to investigate the interfaces and reaction layers at the nanoscale nor the fine-scale microstructure formed as a result of the film deposition. Thus, the objective of this work is to grow ultra-thin AlN films on Si and on GaN/sapphire substrates using PDALD at low temperatures of 360–420 °C and to investigate the interfaces, defects, and fine-scale microstructure of these films.

## 2. Experimental

### 2.1. Growth Parameters

Initially, the substrates were cleaned using hydrogen plasma; then, they were exposed to nitrogen plasma. The process uses 10–30 pulses of hydrogen and 10 pulses of nitrogen plasma. For AlN deposition, in the first step of the cycle, a trimethyl aluminum (TMA) dose was applied for 0.06 s; then, a 10 s purge and a 20 s 75 sccm nitrogen and 10 sccm hydrogen plasma exposure at 300 W were applied. Argon was used as a carrier gas for the TMA step. The growth temperatures ranged from 360 to 420 °C, and the growth mode is layer-by-layer growth. A Williamson pyrometer was used to record the temperature before and after the growth. The rate of growth was in the range of 0.7 to 1.0 Å/cycle, and all growths were on Si and on GaN/sapphire templates grown by MOCVD.

### 2.2. Characterization Techniques: XRD and TEM

A number of high-resolution X-ray diffraction (XRD) scans of the resulting films were made with CuK$_\alpha$ radiation using a Rigaku 18 kW rotating anode generator. Transmission electron microscopy (TEM) was employed to investigate the structure and the crystallinity of the ALD layers using a JEOL 2200-FX analytical transmission electron microscope (JEOL Ltd. Tokyo, Japan) operated at an accelerating voltage of 200 kV. Cross-sectional TEM

samples were made by gluing and mechanically polishing two specimens to a thickness of 25–50 μm. Finally, the samples were thinned using a Gatan ion-mill, PIPS (Gatan Inc., Pleasanton, CA, USA), operated at a voltage of 4 kV for each gun. Digitalmicrograph™ software 3.4 was used to obtain the Fast Fourier transforms (FFTs) from the high-resolution transmission electron microscopic (HRTEM) images.

### 3. Results and Discussion

TEM was employed to investigate the structural characteristics of the AlN/Si film and interfaces. Figure 1a shows the AlN film ≈10 nm thick, which was deposited at 400 °C on (111) Si with ≈50 cycles. The film was observed to be deposited as different layers, exhibiting dark and light contrast (see Figure 1b). HRTEM has been conducted to investigate the nature of the films, interfaces, and defects. Figure 2a is the HRTEM image near the [11-2] zone of Si showing that the film is epitaxial on (111) Si. We have not observed any threading dislocations, although it is deposited at relatively low temperature. However, one could observe faults that appeared as wavy morphology in the HRTEM image. The interface between Si and AlN looks considerably sharp. To analyze the structure of the AlN film, we extracted the fast Fourier transform (FFT) from the film and the substrate (see Figure 2b), and we compared it with the FFT of the substrate in the [11-2] zone (see the inset of Figure 2a). We observe streaking associated with the reflections from the substrate. The streaking in the 111 direction in the FFT results from the faults in the film. The inverse FFT (IFFT) of the film is shown in Figure 2c. TEM analyses show that the film is hexagonal AlN (SG: $P6_3/mmc$ (194)), and the d-spacings of 0.15 nm and 0.19 nm conform to the spacings of (10-12) and (0002) of hexagonal AlN. Usually, the strain effects in the film are dominant close to the substrate. We have measured the d-spacing from HRTEM ≈10 nm away from the substrate/film interface. One could also observe from TEM images that an orientation relationship exists between the ALD-AlN and Si, which is given by $(111)_{Si}$ ∣∣ $(11\text{-}20)_{AlN}$, $(2\text{-}20)_{Si}$ ∣∣ $(0002)_{AlN}$ and $[\text{-}1\text{-}12]_{Si}$ ∣∣ $[1\text{-}100]_{AlN}$.

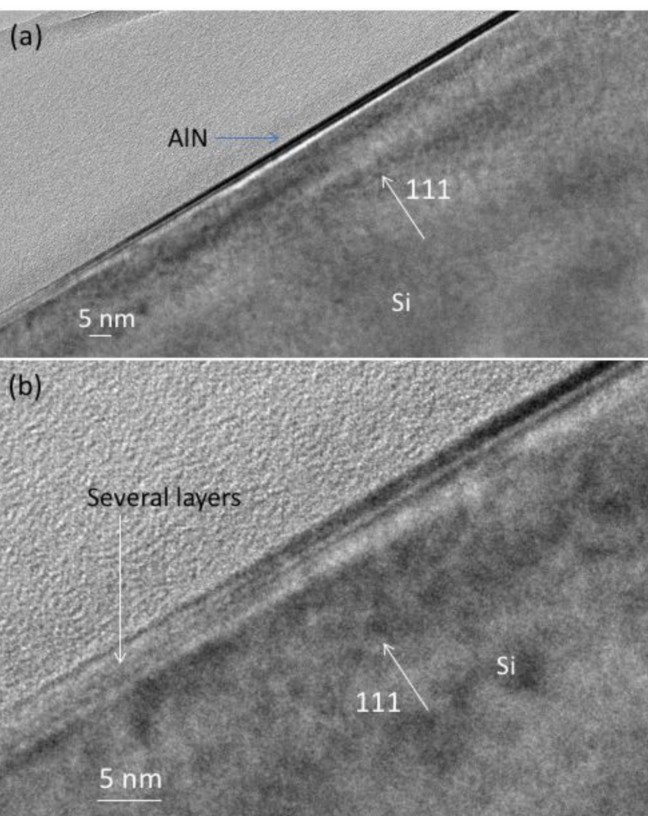

**Figure 1.** (**a**) TEM image showing AlN film on Si (111). (**b**) A higher magnification multibeam image showing the AlN film consisting of several layers.

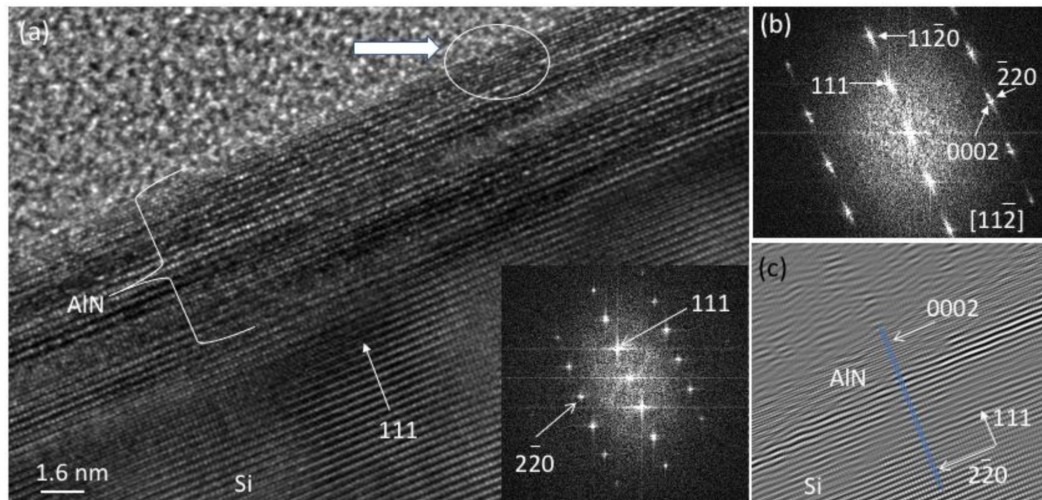

**Figure 2.** (**a**) High-resolution transmission electron microscopic (HRTEM) image near the [11-2] zone of Si showing the AlN layer and relative sharper Si/AlN interface. The Fast Fourier transforms (FFT) from the substrate only has been shown as an inset. (**b**) FFT obtained from the portion of film and the substrate. The spots have been indexed. (**c**) IFFT image of the AlN film and the substrate.

The HRTEM image near the [112] zone (see Figure 3) of Si from another portion of the film at a higher magnification shows a rather sharp interface with a dot-dashed line. The crystallographic a-direction <11-20> is indicated in the image. In the observed orientation relationship, the estimated strain, $(d_{2\text{-}20}{}^{Si}\text{-}d_{0002}{}^{AlN})/d_{2\text{-}20}{}^{Si}$, perpendicular to the growth direction is 0.8%, using $d_{2\text{-}20}{}^{Si}$ = 0.191 nm and $d_{0002}{}^{AlN}$ = 0.1895 nm. The strain type in this case would be tensile. However, for cubic AlN on Si (111), the strain would be more than 10% as $d_{200}{}^{AlN}$ = 0.218 nm, suggesting that the hexagonal AlN would be favored on Si (111) as compared to the cubic AlN on Si (111).

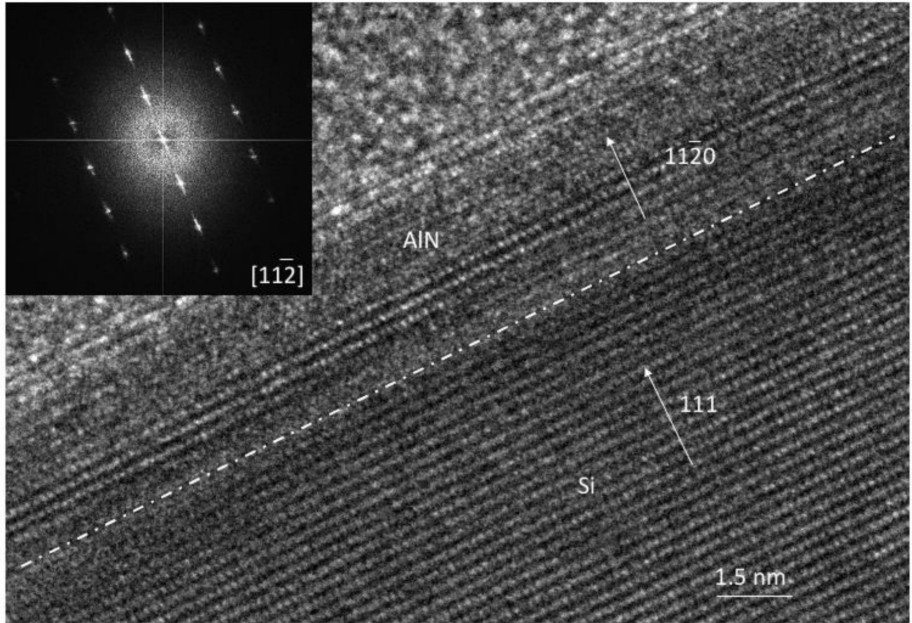

**Figure 3.** HRTEM image showing a rather sharp AlN/Si interface near the [11-2] zone of the substrate. FFT from the substrate and the film has been shown as an inset.

To investigate the crystal structure of the film on GaN/shappire, XRD was used. Initially, we obtain XRD on the GaN/sapphire substrate (see Figure 4). It shows clearly the (0001) oriented sapphire and wurtzitic GaN with (0002) and (0004) peaks. Figure 4 shows

a series of XRD scans on AlN films deposited after a different number of cycles: 50, 100, 250, and 500 cycles, respectively, at 380 °C. A broad peak was observed on the right side of the (0002) GaN diffraction peak. The intensity of this broad peak appeared to increase with the increase in number of cycles. The d-spacing of the broad peak is 0.251 nm, which conforms the d-spacing of 111 zinc blend (cubic) AlN. We observe that the film quality degrades when the deposition temperature is lower. We have employed here grazing incidence small angle X-ray scattering to obtain the optimum growth temperature effects. Our results show that the optimum growth temperature is 420 °C. The increase in growth temperature gives better film; in particular, it lowers the impurity content in the film. The objective here is to grow epitaxial AlN at a lowest possible temperature. We observe that the film is not epitaxial at a temperature of 360 °C in the series [32]. The temperature effects on crystal structure and the transformation kinetics are beyond the scope of the present work. In addition, the grazing incidence small angle X-ray scattering has been used to study the morphology of the surface during AlN growth on (11-20) sapphire by atomic layer deposition [32]. They reported that the films are textured aluminum nitride, and XRD scans show a (1010) peak, indicating wurtzitic (hexagonal) AlN. Note that in the present work, the orientation of sapphire is (0001) and the AlN was deposited on (0001) GaN. This suggests that the orientation of the layer on which it is deposited plays a major role in detecting the crystal structure of AlN.

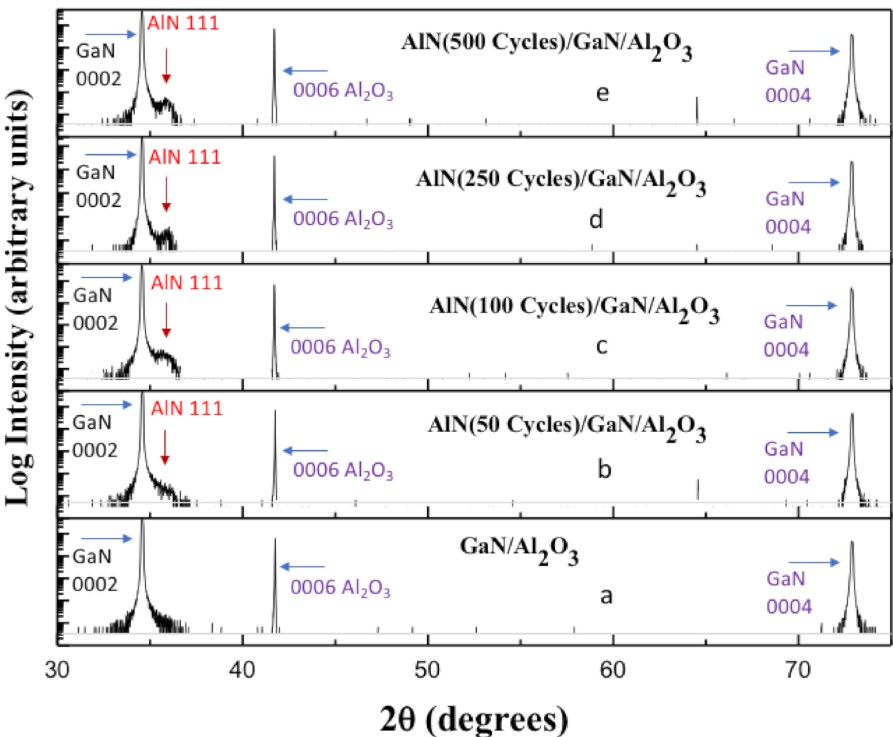

**Figure 4.** A series of XRD patterns in log scale from AlN films on GaN/sapphire as a function of number of reaction cycles at 420 °C. XRD on GaN/sapphire is shown at the bottom panel.

TEM was carried out on these films to further investigate the interfaces and structural characteristics of the AlN films deposited on GaN/sapphire substrate templates. We at first investigate the GaN/sapphire interface. Figure 5a is an HRTEM image showing the GaN on (0001)-oriented sapphire. Threading dislocations were observed to nucleate at the interface. It shows that the (0001) plane of GaN is parallel to the (0001) plane of sapphire. A low-magnification TEM showing GaN and sapphire layers is given as an inset. Figure 5b is the high-magnification HRTEM image of GaN close to the [1-210] zone showing the 0001 (d-spacing = 0.51 nm) and 10-10 lattice fringes. In this zone, the (0001) plane appears as a result of double diffraction. The corresponding fast Fourier transform (FFT) shows the

0001, 0002, 10-10, and 10-11 reflections of wurzitic GaN close to the [1-210] zone (see the index of Figure 5b), which were obtained from the HRTEM image.

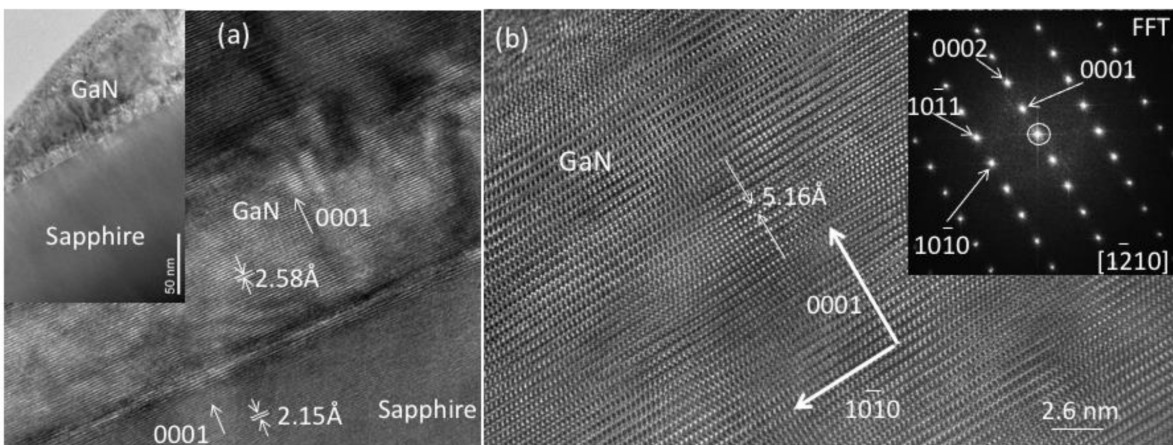

**Figure 5.** (**a**) TEM image showing GaN film on (0001) sapphire. (**b**) HRTEM image showing the lattice (0001) and (10-10) planes close to the [1-210] zone. The FFT from the HRTEM image is shown as an inset.

The AlN film, grown on (0001) GaN, exhibited an orientation relationship, see Figure 6a–e, that can be written as $(0001)_{GaN} \parallel \{111\}_{AlN}$ and $<1210>_{GaN} \parallel <220>_{AlN}$. Figure 6b,c are the FFTs obtained from AlN and GaN layers, respectively. The FFT from the AlN appears complex. The bright spots were indexed as the fcc (zinc blend) AlN close to [110] zone (see Figure 6d,e). In this zone, two sets of {111} planes with spacing 0.251 nm were observed. The angle between 111 spots is 70.5°, suggesting it is indeed a cubic crystal. The extra spots in Figure 6b,d appear most probably due to intermixing during the growth process. Note that the growth process uses the metal–organic precursors and reactive radicals from a plasma source. The intermixing reaction layer, somewhat disordered, is approximately 8 to 10 nm thick (see Figure 7). This layer consists of two phases, cubic AlN and possibly wurtzitic AlGaN. As we go away from the GaN, the fraction of AlN increases. Using density functional theory and the quasi-harmonic approximation to study the phase diagram of AlN, it was shown by Seigel et al. [33] that there is a small energy difference between the wurtzite and zinc blende phases. Due to this small difference in energy, the zinc-blende phase of AlN could be stabilized due to the strain created by the substrate.

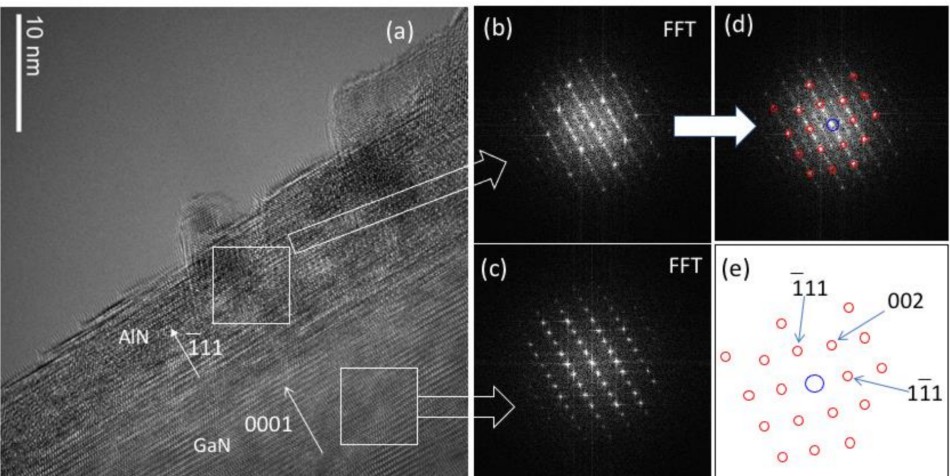

**Figure 6.** (**a**) HRTEM image showing the AlN layer grown on (0001) GaN. (**b**) FFT from AlN layer, (**c**) FFT from GaN, (**d**) FFT showing the encircled spots in [110] zone of zinc blend AlN, (**e**) The indexed pattern in [110] zone.

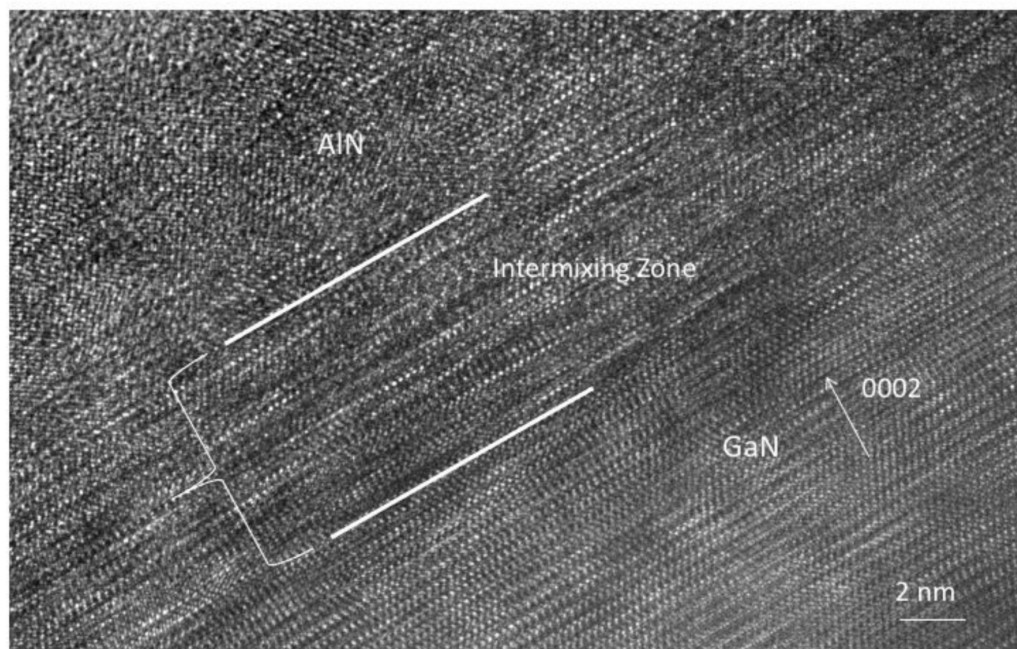

**Figure 7.** HRTEM showing the intermixing at the AlN/GaN interface. The interface layer is approximately 6 to 8 nm thick.

We have estimated the strain in the observed relationship between hexagonal GaN and cubic AlN perpendicular to the growth direction. In the perpendicular direction, the (11-20) of GaN is parallel to (220) AlN. The strain, $(d_{11\text{-}20}^{GaN} - d_{220}^{AlN})/d_{11\text{-}20}^{GaN}$, estimated using the d-spacings of 11-20 of GaN and 220 of AlN, turns out to be 3%. Assuming that the crystal structure of AlN is hexagonal on hexagonal GaN and the (11-20) GaN is parallel to (11-20) of AlN, one could estimate the strain. The estimated strain in this case would be 5% perpendicular to the growth direction. This suggests that the strain would be substantially lowered for cubic–AlN on hexagonal GaN as compared to the hexagonal AlN on hexagonal GaN. In particular, the metastable cubic phase can be stabilized close to the substrate. Note also that we are growing the layer at non-equilibrium conditions, which could induce metastable phases. As you go away from the substrate, the strain effect would be substantially reduced and the stable phase of AlN, hexagonal phase, can nucleate. For the growth on the Si substrate, we observe the stable hexagonal phase on (111) Si, which suggests that the 111 Si surface promotes the hexagonal phase. All these confirm that the strain plays a greater role in dictating the crystal structure upon growth, as the energy difference between hexagonal and cubic AlN is small. More studies are required in the future to optimize the AlN growth at low temperatures, considering all variables and their interactions.

## 4. Summary and Conclusions

In conclusion, we demonstrated the growth of ultra-thin AlN films on Si (111) and on GaN/(0001) sapphire substrate using atomic layer epitaxy at relatively low temperatures. In case of the Si substrate, the AlN film is epitaxial on (111) Si with no threading dislocations. It has deposited with a hexagonal structure with a sharp interface with Si. Similarly, the XRD results showed that the AlN is grown epitaxially with cubic phase on the GaN/(0001) sapphire substrate, which is consistent with the TEM studies. Furthermore, TEM studies showed that the interface is not sharp, containing a transition layer with two phases GaAlN and AlN. The deposited AlN film is stabilized in metastable cubic phase close to the interface. We showed that the nature of the substrate and the resulting the strain at the interface play a major role in dictating the crystal structure of AlN.

**Author Contributions:** Conceptualization, R.G. and C.E.J.; methodology, N.N. and C.E.J.; formal analysis, R.G. and S.Q.; writing—original draft preparation, R.G.; writing—review and editing, C.E.J. and S.Q.; funding acquisition, C.E.J. All authors have read and agreed to the published version of the manuscript.

**Funding:** U.S. Naval Research Laboratory funded this work (the 6.1 research program).

**Institutional Review Board Statement:** Not Applicable.

**Informed Consent Statement:** Not Applicable.

**Data Availability Statement:** Data is contained within the article

**Acknowledgments:** We like to thank the grand challenge program on Ultrathin Multicomponent Electronic Materials.

**Conflicts of Interest:** The authors declare no conflict of interest.

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
