# Peer review of "Microstructure and Interfaces of Ultra-Thin Epitaxial AlN Films Grown by Plasma-Enhanced Atomic Layer Deposition at Relatively Low Temperatures"

_coatings, doi:10.3390/coatings11040482_

Round 1
Reviewer 1 Report
The article is fine. Especially the elaboration of the results and the comprehensive discussion - and especially the in-depth discussion of all the research methods and results from each study - are good impressions.
Few minor observations:
- I suggest that the experimental section should be divided into sub-sections (reagents, apparatus and procedure) otherwise is very difficult to follow.
- Please use consecutive numeration of the manuscript sections (1. Introduction, 2. Experimental, 3….)
Author Response
The article is fine. Especially the elaboration of the results and the comprehensive discussion - and especially the in-depth discussion of all the research methods and results from each study - are good impressions.
Few minor observations:
- I suggest that the experimental section should be divided into sub-sections (reagents, apparatus and procedure) otherwise is very difficult to follow.
Ans: Modified the experimental section.
2. Please use consecutive numeration of the manuscript sections (1. Introduction, 2. Experimental, 3….)
Ans: We have used the consecutive numeration in the manuscript.
Reviewer 2 Report
The authors describe the microstructure and interface of ultrathin epitaxial AlN films. The work is of interest to readers of Coatings magazine. In addition, there are a number of comments on the work.
1. The article uses the PE-ALD method, I propose to correct this in the title.
2. Since the title refers to relatively low growth temperatures, it is necessary to show this in an abstract.
3. Everywhere in the text, especially in the Introduction, the word "relatively" is written, this is not quite a scientific word, I propose to remove and write the actual values of the growth temperatures from the literature cited.
4. At your discretion, I propose to name the article as follows: Microstructure and Interfaces of Ultra-Thin Epitaxial AlN Films Grown by Plasma-Enhanced Atomic Layer Deposition at Low Temperatures.
5. In the Introduction, you should specify other alternative ALD growth technologies for AlN and compare them with your growth values. For example, [Atomic Layer Deposition of Aluminum Nitride Using Tris (diethylamido) aluminum and Hydrazine or Ammonia / DOI: 10.1134 / S1063739718020026], [Low-Temperature Thermal Atomic Layer Deposition of Aluminum Nitride Using Hydrazine as the Nitrogen Source / doi.org/10.3390 / ma13153387].
6. In the experimental part, it is necessary to give the technological growth modes, such as the choice of the temperature window. Weight gain values.
7. Figure 4 shows XRD spectra but cropped at the bottom. I propose to show the whole spectrum.
Author Response
The authors describe the microstructure and interface of ultrathin epitaxial AlN films. The work is of interest to readers of Coatings magazine. In addition, there are a number of comments on the work.
- The article uses the PE-ALD method, I propose to correct this in the title.
Ans: We would like to thank the reviewer for his comments. We have modified the title.
- Since the title refers to relatively low growth temperatures, it is necessary to show this in an abstract.
Ans: We have modified the abstract as suggested by the reviewer.
- Everywhere in the text, especially in the Introduction, the word "relatively" is written, this is not quite a scientific word, I propose to remove and write the actual values of the growth temperatures from the literature cited.
Ans: As suggested by the reviewer, we used the actual growth temperature values in the modified manuscript.
- At your discretion, I propose to name the article as follows: Microstructure and Interfaces of Ultra-Thin Epitaxial AlN Films Grown by Plasma-Enhanced Atomic Layer Deposition at Low Temperatures.
Ans: We thank the reviewer again for proposing the relevant title of the manuscript. We have modified the title according to his suggestion.
- In the Introduction, you should specify other alternative ALD growth technologies for AlN and compare them with your growth values. For example, [Atomic Layer Deposition of Aluminum Nitride Using Tris (diethylamido) aluminum and Hydrazine or Ammonia / DOI: 10.1134 / S1063739718020026], [Low-Temperature Thermal Atomic Layer Deposition of Aluminum Nitride Using Hydrazine as the Nitrogen Source / doi.org/10.3390 / ma13153387].
Ans: We thank the reviewer for this comment. Recently, ALD growth for AlN have been reported in the temperature range of 150–280 °C using Tris (diethylamido) aluminum and Hydrazine (N2H4) or Ammonia [see ref. 29] and in the temperature range of 175-350 °C using trimethyl aluminum (TMA) and hydrazine [see ref. 30]. The use of highly reactive nitrogen source, such as Hydrazine, provides higher deposition rate per cycle as compared to ammonia. The low temperature deposition of AlN, however, shows higher level of impurity content, particularly hydrogen [29].
- In the experimental part, it is necessary to give the technological growth modes, such as the choice of the temperature window. Weight gain values.
Ans: The growth mode is layer by layer growth. We have included this in the experimental section. We have not recorded the weight gain values.
- Figure 4 shows XRD spectra but cropped at the bottom. I propose to show the whole spectrum.
Ans: The XRD spectra is taken in log scale to obtain information from the thin film.
Reviewer 3 Report
This manuscript proposes microstructure and interfaces of ultra-thin epitaxial AlN films grown by atomic layer deposition at relatively low temperatures. The topic is interesting, and certainly consistent with the contents to be proposed to the readers of “Coatings”. However, the manuscript is not so well written and should be improved to be read with pleasure: this represents an important aspect in the current scenario of publications in international journals. Overall, I think that this manuscript could be accepted if the Authors will be able to take into account the following major revisions (in terms of bibliographic updates, grammar corrections and content deepening):
- Detailed revisions: I spent several hours reading this manuscript, and Authors are asked to follow carefully the attached PDF file where I highlighted some points to be addressed. The attached file also contains language mistakes and typos (they are many in this work and should not be present when submitting a manuscript to an international journal: Authors are asked to check the manuscript better next time); some questions related to manuscript contents could also be present and Authors must consider them properly before submitting the revised manuscript. A point-by-point reply is required when the revised files are submitted.
- Considering the amount of mistakes and typos present in this manuscript, a further check carried out by a native English speaker or by a professional English language center is suggested.
- Authors considered different parameters when setting experimental conditions, obtaining the best value for each of them. However, it is known that the optimization of two or more parameters (such as the composition of glasses in this work) is easier, more statistically correct and more functional if performed with a multivariate study (experimental design, chemometrics). Of course, I cannot ask the Authors to re-set from the beginning the planning of experiments according to this way of working. However, the Authors should report in the final part of the manuscript that such a type of study will be done in future works, in order to take into account all the relevant variables and their interactions, finding the best experimental conditions for the proper functioning of their materials/devices/reactors. Concerning this issue, Authors should provide to the readers a few references of works carried out with this multivariate method (design of experiments) in other studies, such as in energy, environmental, sensors, etc.; some of them are suggested in the attached file.
- Authors should provide a clear explanation on the experimental error of the proposed research work. In particular, reproducibility of the phenomena described in the manuscript should be clearly stated in the “Results and Discussion” section; besides, some notes in the “Materials and Methods” section should be added highlighting which kind of experimental approach has been followed to check the reproducibility of the proposed system, the latter being of noteworthy importance in the present research field.
- References: an article submitted to a journal should be consistent with the contents that it typically proposes in its table of contents. However, by checking the references of this manuscript, I did not find any articles published in this journal: this sounds rather strange. Maybe, Authors could check better the topics recently addressed by this journal, studying its table of contents and enriching the Introduction (as mentioned above) with some articles connected to this field.

Author Response
This manuscript proposes microstructure and interfaces of ultra-thin epitaxial AlN films grown by atomic layer deposition at relatively low temperatures. The topic is interesting, and certainly consistent with the contents to be proposed to the readers of “Coatings”. However, the manuscript is not so well written and should be improved to be read with pleasure: this represents an important aspect in the current scenario of publications in international journals. Overall, I think that this manuscript could be accepted if the Authors will be able to take into account the following major revisions (in terms of bibliographic updates, grammar corrections and content deepening):
Ans: Thank you very much for your comments. We have corrected the typos and English.
- I spent several hours reading this manuscript, and Authors are asked to follow carefully the attached PDF file where I highlighted some points to be addressed. The attached file also contains language mistakes and typos (they are many in this work and should not be present when submitting a manuscript to an international journal: Authors are asked to check the manuscript better next time); some questions related to manuscript contents could also be present and Authors must consider them properly before submitting the revised manuscript. A point-by-point reply is required when the revised files are submitted. Considering the amount of mistakes and typos present in this manuscript, a further check carried out by a native English speaker or by a professional English language center is suggested.
Ans: We thank you for your suggestions. We have corrected the typos and language mistakes in the modified manuscript.
- Authors considered different parameters when setting experimental conditions, obtaining the best value for each of them. However, it is known that the optimization of two or more parameters (such as the composition of glasses in this work) is easier, more statistically correct and more functional if performed with a multivariate study (experimental design, chemometrics). Of course, I cannot ask the Authors to re-set from the beginning the planning of experiments according to this way of working. However, the Authors should report in the final part of the manuscript that such a type of study will be done in future works, in order to take into account all the relevant variables and their interactions, finding the best experimental conditions for the proper functioning of their materials/devices/reactors. Concerning this issue, Authors should provide to the readers a few references of works carried out with this multivariate method (design of experiments) in other studies, such as in energy, environmental, sensors, etc.; some of them are suggested in the attached file.
Ans: We would like to thank you for this comment. The objective of this work is to grow AlN films on Si and on GaN/sapphire substrate using PDALD at relatively low temperatures, and to investigate the interfaces, defects and fine scale microstructure of these films. Here we are not very much concerned with optimizing any parameters. As suggested by the reviewer, we stated at the end that such studies can be done in future taking into account all variables and their interactions.
3. Authors should provide a clear explanation on the experimental error of the proposed research work. In particular, reproducibility of the phenomena described in the manuscript should be clearly stated in the “Results and Discussion” section; besides, some notes in the “Materials and Methods” section should be added highlighting which kind of experimental approach has been followed to check the reproducibility of the proposed system, the latter being of noteworthy importance in the present research field.
Ans: We examined number of samples in a given condition using XRD and the results are consistent for each run, suggesting that it is reproducible.
4. References: an article submitted to a journal should be consistent with the contents that it typically proposes in its table of contents. However, by checking the references of this manuscript, I did not find any articles published in this journal: this sounds rather strange. Maybe, Authors could check better the topics recently addressed by this journal, studying its table of contents and enriching the Introduction (as mentioned above) with some articles connected to this field.
Ans: This article is actually intended for a special issue on Epitaxial Thin Films. In addition, we added a reference (see Ref. 33) published in MDPI.
Reviewer 4 Report
To my regard, this review paper about the Microstructure and Interfaces of Ultra-Thin Epitaxial AlN Films Grown by Atomic Layer Deposition at Relatively Low Temperatures is of interest for the scientific community. In my opinion, this work cannot be published in the present form, and should only be considered further. I think that the introduction and conculsions are poor and should be expanded. In addition, the measurement technique used to make the samples and a diagram of the samples made should be better described.
Author Response
To my regard, this review paper about the Microstructure and Interfaces of Ultra-Thin Epitaxial AlN Films Grown by Atomic Layer Deposition at Relatively Low Temperatures is of interest for the scientific community. In my opinion, this work cannot be published in the present form, and should only be considered further. I think that the introduction and conculsions are poor and should be expanded. In addition, the measurement technique used to make the samples and a diagram of the samples made should be better described.
Ans: We would to thank the reviewer for the comments. We have expanded the introduction part by including the recent work done on AlN growth at low temperature and introduced three references (Ref. 29, 30 and 31). For microstructural characterization, we have used transmission electron microscopy (TEM) and X-ray diffraction (XRD) methods, which are very standard methods for thin film characterization. Preparing the cross-section TEM sample is also very common method.
Round 2
Reviewer 2 Report
Thanks for the answers. I propose to accept it for publication.
Reviewer 3 Report
The manuscript has been properly amended and I recommend its publication.